# Heart Failure and Atrial Fibrillation: Diastolic Function Differences Depending on Left Ventricle Ejection Fraction

**DOI:** 10.3390/diagnostics12040839

**Published:** 2022-03-29

**Authors:** Ruxandra-Nicoleta Horodinschi, Camelia Cristina Diaconu

**Affiliations:** 1Department 5, “Carol Davila” University of Medicine and Pharmacy, 050474 Bucharest, Romania; ruxy691@yahoo.com; 2Cardiology Clinic, Clinical Emergency Hospital of Bucharest, 014461 Bucharest, Romania; 3Internal Medicine Clinic, Clinical Emergency Hospital of Bucharest, 014461 Bucharest, Romania

**Keywords:** heart failure, atrial fibrillation, diastolic function

## Abstract

**Background**: Heart failure (HF) and atrial fibrillation (AF) are prevalent cardiovascular diseases, and their association is common. Diastolic dysfunction may be present in patients with AF and all types of HF, leading to elevated intracardiac pressures. **The objective of this study** was to analyze diastolic dysfunction in patients with HF and AF depending on left ventricle ejection fraction (LVEF). **Material and methods**: This prospective study included 324 patients with chronic HF and AF (paroxysmal, persistent, or permanent) hospitalized between January 2018 and March 2021. The inclusion criteria were age older than 18 years, diagnosis of chronic HF and AF, and available echocardiographic data. The exclusion criteria were a suboptimal echocardiographic view, other cardiac rhythms than AF, congenital heart disease, or coronavirus 2 infection. Patients were divided into three subgroups according to LVEF: subgroup 1 included 203 patients with HF with reduced ejection fraction (HFrEF) and AF (62.65%), subgroup 2 included 42 patients with HF with mildly reduced ejection fraction (HFmrEF) and AF (12.96%), and subgroup 3 included 79 patients with HF with preserved ejection fraction (HFpEF) and AF (24.38%). We performed 2D transthoracic echocardiography in all patients. Statistical analysis was performed using R software. **Results:** The E/e′ ratio (*p* = 0.0352, OR 1.9) and left atrial volume index (56.4 mL/m^2^ vs. 53.6 mL/m^2^) were higher in patients with HFrEF than in those with HFpEF. **Conclusions**: Patients with HFrEF and AF had more severe diastolic dysfunction and higher left ventricular filling pressures than those with HFpEF and AF.

## 1. Introduction

Heart failure (HF) and atrial fibrillation (AF) are prevalent cardiovascular diseases, and their association is common. According to the most recent guideline of the European Society of Cardiology, HF is divided into three categories depending on left ventricle ejection fraction (LVEF): HF with reduced LVEF ≤ 40% (HFrEF); HF with mildly reduced LVEF, between 41–49% (HFmrEF); and HF with preserved LVEF ≥ 50% (HFpEF) [1]. Transthoracic echocardiography is the most useful, rapid, and accessible imaging method of diagnosis in patients with HF (acute and chronic) that establishes the type of HF on the basis of LVEF and the cause of HF by assessing structural and functional cardiac anomalies [2].

Diastolic dysfunction may be present in patients with AF and all types of HF, leading to elevated intracardiac pressures [3,4,5]. Invasive measurements of LV stiffness and filling pressure represent the gold standard in diastolic function assessment, but are difficult to perform in daily clinical practice. Therefore, Doppler echocardiography is usually used to evaluate diastolic function through the early diastolic transmitral flow velocity (E wave) to late (atrial) diastolic transmitral flow velocity (A wave) ratio and LV filling pressures using the E wave velocity (determined by pulse Doppler) to e′ velocity (assessed by tissue Doppler) ratio [6]. In patients with HFpEF, diastolic dysfunction represents the etiology of HF that may cause symptoms as severe as systolic dysfunction. Diastolic function assessment represents an essential part of the echocardiographic examination in all patients with HF [6,7]. LV filling pressures determined by noninvasive methods correlate well with the values obtained by invasive methods [8,9]. Diastolic dysfunction often precedes systolic dysfunction [10,11].

AF is the most common sustained supraventricular arrhythmia [12]. HF and LV diastolic dysfunction share common risk factors with AF, such as old age, arterial hypertension, diabetes mellitus, and obesity [13,14].

In patients with AF, the echocardiographic evaluation of diastolic function is more challenging. In these patients, the A wave is lacking because of the absence of atrial contraction. The E wave to mitral annular velocity (e′ wave) ratio is a useful parameter that is used to evaluate LV filling pressures, even in patients with AF [15,16]. Left atrial volume (LAV) is another parameter used to evaluate diastolic dysfunction in patients in sinus rhythm, but in those with AF, the left atrium may be enlarged because of AF and is not a reliable indicator of diastolic dysfunction.

**The objectives of this study** were to analyze the severity of diastolic dysfunction in patients with HF and AF depending on their LVEF and investigate the impact of comorbidities on the severity of diastolic dysfunction.

## 2. Materials and Methods

This was a prospective, observational, case–control study that included 4122 patients consecutively admitted to the Clinical Emergency Hospital of Bucharest, Romania, with the diagnosis of chronic HF and hospitalized between January 2018 and March 2021. Of these patients, we selected 1236 patients who associated paroxysmal, persistent, or permanent AF.

The diagnosis of HF was established on the basis of the European Society of Cardiology guidelines: HFrEF—presence of symptoms and signs of HF and LVEF < 40%; HFmrEF—symptoms and signs of HF and LVEF between 41–49%; HFpEF—symptoms and signs of HF, LVEF ≥ 50% and the presence of cardiac structural or functional abnormalities leading to LV diastolic dysfunction or elevated LV filling pressures, including raised natriuretic peptides (brain natriuretic peptide > 105 pg/mL and amino terminal prohormone of brain natriuretic peptide > 365 pg/mL, taking into account that all patients included in the study had AF) [1].

The diagnosis of AF was established using a standard 12-lead electrocardiogram or automated continuous monitoring of cardiac rhythm for 24–72 h. The inclusion criteria were age older than 18 years, concomitant diagnosis of chronic HF and AF, complete echocardiographic data available, signed informed consent to participate in this study. We included in the study only the patients who had AF at the moment of inclusion in the study (concurrently with the moment when the echocardiography was performed, independently if AF was diagnosed at inclusion or before). Most patients were already diagnosed with AF (permanent, persistent, or recurrent paroxysmal) before inclusion in the study. An ECG was performed at inclusion in all patients and 24 h ECG Holter study was performed only in some patients with paroxysmal AF. In patients with persistent or permanent AF, 24 h ECG Holter study was performed only in those with uncontrolled heart rate or in those suspected to associate other atrial or ventricular arrhythmias. Patients with a history of AF who were in another cardiac rhythm at the initial assessment were excluded from the study. The exclusion criteria were an incomplete echocardiographic report, suboptimal echocardiographic view leading to the impossibility to obtain all the necessary data required in the protocol of this study, other cardiac rhythms than AF, congenital heart disease, coronavirus 2 infection (for patients recruited between March 2020 and March 2021).

After applying the inclusion and exclusion criteria, 324 patients remained in the study. The patients were divided into three subgroups, according to LVEF: subgroup 1 included 203 (62.65%) patients with HFrEF and AF, subgroup 2 included 42 (12.96%) patients with HFmrEF and AF, subgroup 3 included 79 (24.38%) patients with HFpEF and AF.

Permanent AF was the most frequent type and it was present in 61.1% of the patients (198 patients: 133 patients in subgroup 1, 20 patients in subgroup 2, 45 patients in subgroup 3). Persistent AF was present in 24.07% of the patients (78 patients: 40 in subgroup 1, 8 in subgroup 2, 30 in subgroup 3), and paroxysmal AF was present in 14.8% of the patients (48 patients: 30 in subgroup 1, 14 in subgroup 2, 4 in subgroup 3). The majority of the patients with the diagnosis of chronic HF and AF were excluded from the study mainly because of the impossibility to obtain their initial echocardiography reports or because they had a suboptimal ultrasound view; other reasons for excluding patients were the presence of other cardiac rhythms, such as atrial flutter or multifocal atrial tachycardia; the lack of signed informed consent to participate in the study; and SARS-CoV-2 infection for patients included between March 2020 and March 2021. The study protocol is shown is Figure 1.

Baseline demographics were obtained at inclusion in the study. The following data were collected: general data (sex, age), medical history, comorbidities, current disease status, reason for current hospitalization, current treatment.

The study respected the ethical standards of the Helsinki Declaration of 1975, as revised in 2008(5), as well as the national law. We ensured that patients’ rights were protected and the confidentiality of their data was maintained. The study was approved by the Ethics Committee of the Clinical Emergency Hospital of Bucharest, Romania (approval number 4714/24.05.2019).


**Laboratory tests**


In order to completely assess the current status of cardiovascular disease and comorbidities such as coronary artery disease, chronic kidney disease (CKD), diabetes mellitus, dyslipidemia, and liver disease, blood tests were performed in all patients included in the study using an ABX Pentra XL 80 hematology autoanalyzer for complete cell blood count (leukocytes 4000–9000/µL; haemoglobin 12.6–17.2 g/dL; platelets 150.000–350.000/µL) and a BioMajesty biochemistry autoanalyzer for creatinine (0.7–1.4 mg/dL), blood urea nitrogen (19–43 mg/dL), serum sodium (137–145 mmol/L), serum potassium (3.5–5 mmol/L), aminotransferases (aspartate aminotransferase 14–50 U/L, alanine aminotransferase 10–50 U/L), glycaemia (75–110 mg/dL), troponin I (<5 ng/mL), creatine kinase (55–170 U/L), creatine kinase-MB (10–16 U/L), total cholesterol (140–200 mg/dL), triglycerides (30–150 mg/dL).


**Transthoracic echocardiography**


In all patients enrolled in the study, 2D transthoracic echocardiography was performed either at inclusion in the study or the data was extracted from the medical records of the patients (only if the data were consistent with the study protocol of examination and if the patients had AF at the moment echocardiography was performed).

Commercially available ultrasound systems, such as Philips CX 50 or Vivid 9 machine, were used to examine the patients. Conventional measures, such as the dimensions of the LV walls, LV end-diastolic and end-systolic diameters and volumes, LA diameter and volume, right atrial diameter and area, right ventricular diameter, were obtained. Left atrium and ventricle volumes were indexed to body surface area. LVEF was calculated in the apical four- and two-chambers views using the modified Simpson’s biplane method at inclusion in the study. On the basis of this, patients were classified into three subgroups, namely, with HFrEF and AF, HFmrEF and AF, HFpEF and AF. E wave was measured in apical 4-chamber view using pulse-Doppler imaging. Tissue Doppler was used to determine average e′ (the average between e′ lateral and e′ septal). Diastolic function was assessed by pulse Doppler used to determine E wave velocity and tissue Doppler for e′ wave velocity. Right ventricle function was assessed using tricuspid annular plane systolic excursion and tricuspid annular peak systolic velocity (s′) obtained by tissue doppler. Valvular pathologies were assessed using color, pulse, and continuous Doppler. Aorta and pericardium were also evaluated.


**Statistical analysis**


Statistical analysis was performed using R software version 4.0.2 (22 June 2020), R Core Team 2020 (R: a language and environment for statistical computing; R Foundation for Statistical Computing, Vienna, Austria). Descriptive statistics were presented as absolute frequencies, mean values ± standard deviation, medians with interquartile range. To evaluate the differences in the E/e′ ratio between the three subgroups of patients, a univariate binary logistic regression, simple or multiple, was used. The dependent variable (“output”) was the presence or absence of a pathologic E/e′ ratio (>15). The independent variables (“input”) were the individual characteristics of the patients, including demographic parameters (age, sex), LVEF, and comorbidities.

Then, the predictors of increased E/e′ ratio in the univariate analysis were selected for multiple binary logistic regression to identify the factors independently associated with elevated E/e′ ratio. A one-way analysis of variance (ANOVA) was performed to determine the differences in the LV end-diastolic volume (LVEDV) between the three subgroups. Then, we applied a post hoc ANOVA analysis to identify the reason for LVEDV differences between the subgroups, using a series of bidirectional Welch *t* tests that compared the subgroups between them. The multiple tests were corrected using the Benjamini–Yekutieli procedure to control the false discovery rate under arbitrary dependence assumptions. A value of *p* < 0.05 was considered statistically significant for all tests.

## 3. Results

The baseline characteristics of patients with HF and AF, according to their LVEF, are shown in Table 1. More than half of the patients (62.65%) had HFrEF, 12.96% had HFmrEF, and 24.38% had HFpEF. Patients with HFpEF were older than those with HFmrEF or HFrEF, and patients with HFmrEF were older than those with HFrEF. Patients with HFrEF were more frequently males, while patients with HFpEF were more frequently females. Obesity, smoking, alcohol intake, and thyroid disorders are risk factors for AF. Obesity was encountered in 109 (33.64%) patients (61 patients with HFrEF and AF, 17 patients with HFmrEF and AF, and 31 patients with HFpEF and AF). Of the patients, 48% were smokers (158 patients, of whom 80 with HFrEF and AF, 19 with HFmrEF and AF, and 59 with HFpEF and AF). Chronic alcohol consumption was found in 88 (27.16%) patients (46 patients with HFrEF and AF, 12 with HFmrEF and AF, and 30 with HFpEF and AF). Thyroid disorders were found in 38 (11.72%) patients (19 with HFrEF and AF, 8 with HFmrEF and AF, and 11 with HFpEF), who were referred to an endocrinologist.

LAVi was higher in patients with HFrEF than in those with HFpEF or HFmrEF. Also, LAVi was higher in patients with HFpEF than in those with HFmrEF. Taking into account the relatively small number of patients with HFmrEF included in the study, a possible explanation of smaller LAVi in this subgroup compared with the subgroup of patients with HFpEF is that less than half of the patients with HFmrEF had permanent AF (47.61%); the other 52.39% of the patients with HFmrEF had paroxysmal or persistent AF, and the restoration of sinus rhythm may have limited left atrial enlargement and dysfunction. Patients with HFrEF and AF had higher E/e′ ratios compared with patients with HFmrEF or HFpEF and AF, while in patients with HFmrEF or HFpEF, the mean E/e′ ratio was similar (Table 2). Applying a univariate binary simple logistic regression, the predictors for an increased E/e′ ratio were HFrEF (1.9 times increase compared with HFpEF) and chronic kidney disease (CKD; 2.06 increase compared with patients with normal kidney function; Table 3).

A multiple logistic regression was used to determine the independent predictors for an increased E/e′ ratio, with backward selection. The independent predictors for an increased E/e′ ratio were HFrEF (an increase by 1.85 times compared with patients with HFpEF) and CKD (an increase of two times compared with patients with normal kidney function) (Table 4).

An analysis of variance (ANOVA) was used to demonstrate that the difference in LVEDVi values was statistically significant (*p* < 0.0001) (Table 5). Moreover, all the differences between the three subgroups were statistically significant (*p* < 0.0001), according to a post hoc analysis with the Benjamini–Yekutieli procedure (Table 6).

## 4. Discussion

The primary endpoint of this study was to determine the differences between the severity of diastolic dysfunction in patients with HF and AF, according to LVEF. To achieve the primary endpoint, we analyzed mainly two echocardiographic parameters, E/e′ ratio and LAVi, that are reliable in patients with AF in whom diastolic function assessment is more challenging.

The cut-off values for LV diastolic dysfunction indices that were considered abnormal were LAVi > 34 mL/m^2^, peak E velocity > 50 cm/s, E/e′ > 15, isovolumic relaxation time ≤ 65 ms, and deceleration time of E wave ≤ 150 ms [6,17,18,19,20,21,22,23,24].

The E/e′ ratio is the most reliable and easy-to-evaluate diastolic parameter that correlates well with LV filling pressures measured by invasive pulmonary catheterism [19,25]. The E/e′ ratio value increases with increases in mean pulmonary capillary wedge pressure [26]. The E value reflects the mitral valve status determined by LV relaxation, and the e′ value reflects the LV walls’ relaxation [27]. When myocardial relaxation is affected, E velocity increases due to mitral valve opening delay and e′ velocity is reduced and delayed [7,27,28]. Transmitral inflow velocity E is directly proportional to the ratio between LA pressure and the relaxation time constant tau, while e′ is inversely proportionate to the tau constant. Therefore, the E/e′ ratio is directly proportional to LA pressure [8]. Although some studies observed that the correlation was stronger in patients with HFrEF, the E/e′ ratio may be used in patients with HFpEF [8,29,30]. Usually, it is preferable to use several echocardiographic indices to appreciate diastolic function, but, despite this, the E/e′ ratio is often used by itself to quantify LV diastolic function [31].

To achieve the primary endpoint, the E/e′ ratio was comparatively evaluated in three subgroups of patients as an indicator of diastolic dysfunction and increased LV filling pressures. The results of this study showed that the E/e′ ratio was higher in patients with HFrEF compared with those with HFpEF, revealing that patients with HFrEF had higher LV filling pressures than those with HFpEF.

The second echocardiographic parameter evaluated in this study was LAVi. LAVi was higher in patients with HFrEF compared with those with HFpEF. LAVi was higher in patients with HFpEF compared with those with HFmrEF. LA volume was higher in patients with HFrEF compared with those with HFpEF in other studies [32,33,34,35,36]. LA remodelling is based on structural (enlargement, fibrosis) and functional (mechanical and electrical dysfunction) disorders that may occur as a result of pressure or volume overload or tachycardia [37,38]. LA dysfunction contributes to HF symptoms’ severity and progression [39,40,41,42,43]. LAV is an important outcome predictor in patients with HF, and the LA functional index represents a strong independent predictor of long-term survival in patients with stable, chronic HF [44,45]. LA abnormalities have an important role in the pathophysiology and progression of LV diastolic dysfunction because LA function is important for maintaining optimal cardiac performance [46,47,48,49]. LA is similar to an elastic reservoir of blood, modulating LV filling, whereas LV influences LA function through the cardiac cycle. Diastolic dysfunction leads to higher filling pressures that are necessary to achieve an adequate filling [50]. Increased LA volume is a mark of long-standing elevated LV filling pressures, and LA volume is more sensitive than LA diameter [8,51,52]. There are several mechanisms that lead to LA dysfunction in patients with HF. One mechanism is the chronic exposure to increased LA preload and afterload. Due to increased preload, at first, LA contractility increases as a compensatory mechanism; then, it begins to decline because of LA remodeling, apoptosis of atrial cardiomyocytes, and collagen matrix increased turnover leading to interstitial fibrosis [53,54]. Atrial fibrosis and the disruption of electrical connections between myocytes lead to atrial refractory period shortening and conduction abnormalities that predispose individuals to electrical ectopic activity and AF development [37]. AF per se causes LA remodeling and fibrosis [37]. The loss of normal atrial electrical activity leads to inefficient atrial contraction [50]. All these pathophysiological changes in patients with HF and AF lead to increased LA volume and stiffness, which are more pronounced compared with patients with HF or AF alone. In patients with HF, without LA enlargement (LA volume less than 34 mL/m^2^), HF is recently developed and filling pressures have not increased [55]. Therefore, LA size and function assessment represents an important part of the echocardiographic examination, in order to evaluate diastolic function in all patients with HF [56].

The study realized by Melenovsky et al. included 198 patients with HF, of whom 51% had HFpEF and 49% had HFrEF, and 40 controls without HF [50]. In all patients, they performed echocardiography, catheterization, and follow-up. All patients with HF had more dilated and dysfunctional LA than controls [50]. Patients with HFpEF had increased LA stiffness, while those with HFrEF had more eccentric LA remodeling [50]. LA enlargement and dysfunction were more severe in patients with HFrEF than in those with HFpEF at similar LA mean pressure, but in contrast, in patients with HFpEF, maximal LA pressure was higher and minimal LA pressure was lower [50]. LA stiffness was greater in patients with HFpEF [50]. AF was associated with increased LA stiffness and more severe LA enlargement, especially in patients with HFpEF [50]. LA dilation and dysfunction were associated with a high risk of mortality only in patients with HFpEF [50].

The secondary endpoint of our study was to determine if there are other factors or comorbidities that influence the severity of diastolic dysfunction in patients with HF and AF. The results revealed that patients with HF, AF, and CKD had higher E/e′ ratios compared with patients with HF, AF, and normal renal function, regardless of LVEF. Diastolic dysfunction is common in patients with chronic kidney disease (CKD), especially in those with advanced CKD, but it may develop in early stages [23]. Cardiovascular risk increases with declines in renal function [23]. Usually, patients with CKD have a profibrotic status because of increased accumulation of collagen in the connective tissue and activation of the renin–angiotensin–aldosterone system, which predispose individuals to myocardial fibrosis [23]. Uremic cardiomyopathy starts with diastolic dysfunction and then evolves with progressive myocardial fibrosis and LV hypertrophy [23]. Arterial hypertension is common in patients with CKD because of fluid retention and excessive arterial stiffness, and it contributes to LV hypertrophy and diastolic dysfunction [27]. Early diastolic dysfunction may be asymptomatic, but it is important to perform a screening echocardiography to assess diastolic function in patients with CKD, regardless of the level of renal function decline, to prevent irreversible cardiac remodeling. Diastolic dysfunction is associated with a higher risk of mortality compared with systolic dysfunction, and it is preferable to diagnose it as early as possible [57,58]. The E/e′ ratio is an accurate parameter in assessing diastolic function in patients with CKD [24]. In these patients, fluid overload with sodium and water retention leads to an increase in E wave velocity, while LV increased stiffness leads to an e′ velocity decrease that is not affected by patients’ fluid status [23]. The study realized by Kim et al. included 136 patients with CKD, of whom 38.9% had systolic dysfunction, and evaluated the impact of diastolic dysfunction, assessed mainly through the E/e′ ratio, on mortality and cardiovascular events [27]. According to this study, an increase in the E/e′ ratio may be associated with an increase in mortality risk and cardiovascular disease incidence in patients with renal impairment, regardless of CKD stage [27].

The results of this study showed that diastolic function worsens with decreases in LVEF in patients with HF and AF and is more severe in patients with HFrEF than in those with HFpEF or HFmrEF. Moreover, we evaluated the impact of different comorbidities on diastolic dysfunction severity, and the results revealed that CKD may influence diastolic function in patients with HF with all ranges of LVEF and AF. Patients with HF, AF, and CKD had more severe diastolic dysfunction compared with patients with HF, AF, and normal renal function.

A limitation of this study was the relatively low number of patients with HFmrEF compared with those with HFrEF or HFpEF. CKD was also present in a small number of patients. There were possible differences in appreciating the ultrasound parameters because of different examiners who performed the transthoracic echocardiography. Other echocardiographic parameters were analyzed, including deceleration time of the E wave, isovolumic relaxation time, and tricuspid regurgitation velocity, but no statistically significant differences between the three subgroups of patients were obtained.

Future research may be focused on evaluating if an early diagnosis of diastolic dysfunction limits its progression to severe stages and if its progression has any link with systolic function decline. Another area of research may be focused on evaluating if the diagnosis of diastolic dysfunction in patients with early-stage renal dysfunction limits CKD progression.

## 5. Conclusions

Diastolic dysfunction is common in patients with HF and AF. Moreover, diastolic dysfunction represents one cause of HFpEF. The E/e′ ratio represents a reliable parameter in the noninvasive evaluation of LV filling pressures, and it can be used in patients with AF. This study demonstrated that patients with HFrEF and AF had higher values for the E/e′ ratio and LAVi compared with patients with HFpEF or HFmrEF and AF. The E/e′ ratio was similar in patients with HFpEF or HFmrEF and AF. Furthermore, we determined that patients with CKD had higher E/e′ ratios compared with those with normal renal function.

## Figures and Tables

**Figure 1 diagnostics-12-00839-f001:**
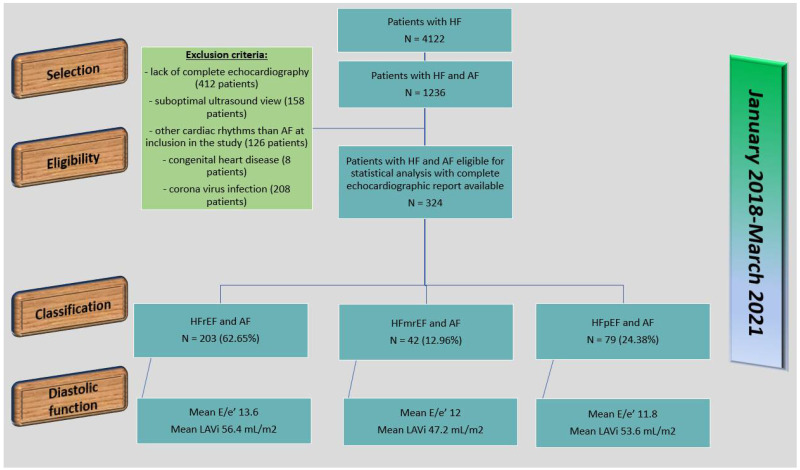
Protocol of the study. *Legend*: HF—heart failure; AF—atrial fibrillation; HFrEF—heart failure with reduced ejection fraction; HFmrEF—heart failure with mildly reduced ejection fraction; HFpEF—heart failure with preserved ejection fraction; LAVi—left atrial volume index.

**Table 1 diagnostics-12-00839-t001:** Baseline characteristics of patients with HF and AF, according to LVEF.

	HFrEF	HFmrEF	HFpEF	All
	(N = 203)	(N = 42)	(N = 79)	(N = 324)
**Age**				
Mean (SD)	70.5 (11.9)	75.0 (10.4)	78.6 (8.85)	73.1 (11.6)
Median (Min, Max)	71.0 (38.0, 100)	75.0 (52.0, 94.0)	79.0 (60.0, 96.0)	73.0 (38.0, 100)
**Sex**				
F	63 (31.0%)	20 (47.6%)	47 (59.5%)	130 (40.1%)
M	140 (69.0%)	22 (52.4%)	32 (40.5%)	194 (59.9%)
**LVEF**				
Mean (SD)	23.9 (7.73)	42.4 (2.60)	52.6 (2.77)	33.3 (14.1)
Median	25.0	40.0	52.0	30.0
(Min, Max)	(5.00, 38.0)	(40.0, 47.0)	(50.0, 60.0)	(5.00, 60.0)
**LAVi**				
Mean (SD)	56.4 (23.9)	47.2 (19.8)	53.6 (23.0)	54.5 (23.3)
Median	52.0	43.0	49.0	50.0
(Min, Max)	(14.0, 153)	(21.0, 100)	(18.0, 134)	(14.0, 153)
**LVEDVi**				
Mean (SD)	89.3 (26.2)	72.6 (15.6)	56.8 (14.8)	79.2 (26.5)
Median (Min, Max)	88.0 (29.0, 198)	71.0 (48.0, 113)	54.0 (19.0, 93.0)	78.0 (19.0, 198)
Missing	1 (0.5%)	0 (0%)	0 (0%)	1 (0.3%)
**CAD**				
No	110 (54.2%)	17 (40.5%)	56 (70.9%)	183 (56.5%)
Yes	93 (45.8%)	25 (59.5%)	23 (29.1%)	141 (43.5%)
**CKD**				
No	141 (69.5%)	33 (78.6%)	59 (74.7%)	233 (71.9%)
Yes	62 (30.5%)	9 (21.4%)	20 (25.3%)	91 (28.1%)
**HT**				
No	67 (33.0%)	13 (31.0%)	21 (26.6%)	101 (31.2%)
Yes	136 (67.0%)	29 (69.0%)	58 (73.4%)	223 (68.8%)

*Legend*: HFrEF—heart failure with reduced ejection fraction; HFmrEF—heart failure with mildly reduced ejection fraction; HFpEF—heart failure with preserved ejection fraction; LAVi—left atrial volume index; SD—standard deviation; LVEDVi—left ventricle end-diastolic volume index; CAD—coronary artery disease; CKD—chronic kidney disease; HT—hypertension; Min—minimum; Max—maximum.

**Table 2 diagnostics-12-00839-t002:** E/e′ ratio in patients with HF and AF, according to LVEF.

	Group A (N = 203)	Group B (N = 42)	Group C (N = 79)	Overall (N = 324)
**E/e′**				
Mean (SD)	13.6 (5.58)	12.0 (6.13)	11.8 (4.58)	12.9 (5.48)
Median (Min, Max)	11.0 (3.60, 36.0)	9.00 (6.51, 34.0)	10.0 (5.70, 25.0)	10.0 (3.60, 36.0)

*Legend*: SD—standard deviation; Min—minimum; Max—maximum.

**Table 3 diagnostics-12-00839-t003:** Predictors for increased E/e′ ratio in patients with HF and AF (univariate binary simple logistic regression).

Parameter		Coefficient	*p*-Value	OR (CI 95%)
**Age**		−0.002	0.7980	0.99 (0.97 to 1.01)
**Sex**	**F**	REFERENCE	-	-
	**M**	−0.02	0.9070	0.97 (0.60 to 1.57)
**HFpEF**		REFERENCE	-	-
**HFmrEF**		0.05	0.8988	1.05 (0.42 to 2.52)
**HFrEF**		1.90	0.0352	1.90 (1.06 to 3.53)
**LAVi**		0.009	0.0734	1.01 (0.99 to 1.02)
**CAD**	**No**	REFERENCE	-	-
	**Yes**	−0.11	0.6730	0.89 (0.55 to 1.43)
**CKD**	**No**	REFERENCE	-	-
	**Yes**	0.72	0.0049	2.06 (1.24 to 3.43)
**HT**	**No**	REFERENCE	-	-
	**Yes**	−0.29	0.2430	0.74 (0.45 to 1.29)
**DM**	**No**	REFERENCE	-	-
	**Yes**	0.12	0.6170	1.13 (0.68 to 1.86)

*Legend*: OR—odds ratio; CI—confidence interval; HFrEF—heart failure with reduced ejection fraction; HFmrEF—heart failure with mildly reduced ejection fraction; HFpEF—heart failure with preserved ejection fraction; LAVi—left atrial volume index; CAD—coronary artery disease; CKD—chronic kidney disease; HT—hypertension; DM—diabetes mellitus.

**Table 4 diagnostics-12-00839-t004:** Independent predictors for an increased E/e′ ratio in patients with HF and AF.

Parameter		Coefficient	*p*-Value	OR (CI 95%)
**HFpEF**		REFERENCE	-	-
**HFmrEF**		0.08	0.8462	1.09 (0.43 to 2.63)
**HFrEF**		0.61	0.0449	1.85 (1.02 to 3.47)
**CKD**	**No**	REFERENCE	-	-
	**Yes**	0.69	0.0076	2.00 (1.20 to 3.47)

*Legend*: OR—odds ratio; CI—confidence interval; HFrEF—heart failure with reduced ejection fraction; HFmrEF—heart failure with mildly reduced ejection fraction; HFpEF—heart failure with preserved ejection fraction; CKD—chronic kidney disease.

**Table 5 diagnostics-12-00839-t005:** Analysis of variance (ANOVA) for LVEDVi in patients with HF and AF.

Source	Df	Sum Sq	Mean Sq	F Stat.	*p*-Value
Group	2	61,802	30,900.80	59.83	<0.0001
Residuals	320	164,990	515.60	-	-

*Legend*: Df—difference; Sum Sq—sum of squares; Mean Sq—mean squared error.

**Table 6 diagnostics-12-00839-t006:** Post hoc analysis with Benjamini–Yekutieli procedure for LVEDVi in patients with HF and AF.

Contrast	Adjusted *p*-Value
HFrEF vs. HFmrEF	<0.0001
HFrEF vs. HFpEF	<0.0001
HFmrEF vs. HFpEF	0.0005

*Legend*: HFrEF—heart failure with reduced ejection fraction; HFmrEF—heart failure with mildly reduced ejection fraction; HFpEF—heart failure with preserved ejection fraction.

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
