# Peer review of "Heart Failure and Atrial Fibrillation: Diastolic Function Differences Depending on Left Ventricle Ejection Fraction"

_diagnostics, 2022, doi:10.3390/diagnostics12040839_

Round 1

Reviewer 1 Report

The authors have tried to address my comments and have made an effort to improve the manuscript.

Author Response

Journal: Diagnostics

Manuscript ID:  diagnostics-1635959

Title: Heart failure and atrial fibrillation: diastolic function differences depending on left ventricle ejection fraction

Dear Academic Editors,

We are very thankful for your quick response and for your comments about our manuscript.

Cordially yours,

Camelia Diaconu, MD, PhD, corresponding author

Reviewer 2 Report

Reviewer’s Comment for Diagnostics

Figure 1 and 2 were too redundant and please consider to remove them or keep only 1 of them.

Among 4122 HF patients, the prevalence of AF was 1236? This prevalence of AF was too low. Any explanation for this? The final analyzed patient population was 324 and even much lower. How many were excluded in each step of exclusion and why? An example flowchart or diagram illustrating the initial and final enrollment or eligible number of patients may be helpful. I think 324 indicated those with available data with also echo available. If that is the case, please detailed these in a more comprehensive manner in Figure 3.

To me, natriuretic peptides (NP) and cutoffs used in current study including brain natriuretic peptide (> 105 pg/mL) and amino terminal prohormone 94 of brain natriuretic peptide (> 365 pg/mL), taking into accounts that all patients included in 95 the study had AF) seems to low. By using ESC 2021 HF criteria and guideline-based NP definition, BNP ≥ 100 pg/mL or NT-proBNP ≥300 pg/mL was for acute HF. In AF patients manifesting acute HF, the level should be much higher.

Was LVEF measured by Biplane Simpson method? The mean and SD of LVEF in HFmrEF and HFpEF were 42.4 (2.60) and 52.6 (2.77), respectively. As we can see, the mean values minus only 1 SD in HFmrEF and HFpEF would be less than 40% and 50%, respectively. There should be more cases fall below 40% and 50% in full spectrum of HFmrEF and HFpEF in current study. This should be explained in detail and seeming unreasonable.

While we are acknowledged that AF was diagnosed through ECG and Holter study, the question would be whether AF was diagnosed and established through medical history or on-site during admission (acute HF phase)? We know that AF may happen during decompensated HF and therefore the prevalence of AF may be even higher in acute HF. 1236 is relatively small in ratio if AF was diagnosed in acute setting. And if the AF in current work was not defined by using admission timepoint ECG and was defined from medical history, then how the echo data regarding structure and function correlated with AF rhythm? Were echo done at the same time of AF diagnosis prior to this admission? And what about the time window?  

Author Response

Journal: Diagnostics

Manuscript ID:  diagnostics-1635959

Title: Heart failure and atrial fibrillation: diastolic function differences depending on left ventricle ejection fraction

Dear Academic Editors,

We are very thankful to you for the pertinent notes; we have carefully read the comments and have revised/ completed the manuscript accordingly. Our responses are given in a point-by point manner below; all the changes to the manuscript are highlighted in red and marked by Track changes.

We hope that in this new form, the manuscript will be suitable for publication.  We did our best to fulfill the expectations and we hope that you will be satisfied with our corrections.

  1. Figure 1 and 2 were too redundant and please consider to remove them or keep only 1 of them.

Response: thank you for your valuable suggestion. We removed figure1 and 2.

  1. Among 4122 HF patients, the prevalence of AF was 1236? This prevalence of AF was too low. Any explanation for this? The final analyzed patient population was 324 and even much lower. How many were excluded in each step of exclusion and why? An example flowchart or diagram illustrating the initial and final enrollment or eligible number of patients may be helpful. I think 324 indicated those with available data with also echo available. If that is the case, please detailed these in a more comprehensive manner in Figure 3.

Response: We included in the study only the patients with HF who were in AF at the inclusion in the study.We excluded patients with history of paroxysmal or persistent AF who were in sinus rhythm or in other cardiac rhythm, such atrial flutter, at inclusion. Indeed, only the patients for whom we hadthe complete echocardiographic report remained in the study, respectively 324; we excluded patients without echocardiographic dataand also those who had only a screening echocardiography or those who had incomplete echocardiographic data. We presented in detail in Figure 3 (protocol of the study, that is now Figure 1 in the manuscript, because we removed Figure 1 and 2) how many patients were excluded according to each exclusion criteria. Only the patients for whom we had a complete echocardiographic report remained in the study for statistical analysis.

  1. To me, natriuretic peptides (NP) and cutoffs used in current study including brain natriuretic peptide (> 105 pg/mL) and amino terminal prohormone 94 of brain natriuretic peptide (> 365 pg/mL), taking into accounts that all patients included in the study had AF seems to low. By using ESC 2021 HF criteria and guideline-based NP definition, BNP ≥ 100 pg/mL or NT-proBNP ≥300 pg/mL was for acute HF. In AF patients manifesting acute HF, the level should be much higher.

Response: Thank you for your observations. We used the cut-off values of the natriuretic peptides that are mentioned inthe ESC Guidelines for the management of patients with HF since 2021 in Table 9, respectively the cut-off values for natriuretic peptides in patients with HF and AF are BNP > 105 pg/mL and NT-proBNP> 365 pg/mL.These values are used only for patients with HFpEF and AF; for patients with HFrEF or HFmrEF and AF, natriuretic peptides are not necessary for the diagnosis.

  1. Was LVEF measured by Biplane Simpson method? The mean and SD of LVEF in HFmrEF and HFpEF were 42.4 (2.60) and 52.6 (2.77), respectively. As we can see, the mean values minus only 1 SD in HFmrEF and HFpEF would be less than 40% and 50%, respectively. There should be more cases fall below 40% and 50% in full spectrum of HFmrEF and HFpEF in current study. This should be explained in detail and seeming unreasonable.

Response: We mentioned in the manuscript that Simpson’s biplane method was performed in all patients.

“LVEF was calculated in the apical four- and two-chambers views using the modified Simpson’s biplane method at inclusion in the study. Based on this, the patients were classified in the three subgroups, respectively with HFrEF and AF, HFmrEF and AF, HFpEF and AF.”

We took into account the value of the LVEF at inclusion in the study.Based on that, we calculated the mean and SD values of the LVEF.

  1. While we are acknowledged that AF was diagnosed through ECG and Holter study, the question would be whether AF was diagnosed and established through medical history or on-site during admission (acute HF phase)? We know that AF may happen during decompensated HF and therefore the prevalence of AF may be even higher in acute HF. 1236 is relatively small in ratio if AF was diagnosed in acute setting. And if the AF in current work was not defined by using admission timepoint ECG and was defined from medical history, then how the echo data regarding structure and function correlated with AF rhythm? Were echo done at the same time of AF diagnosis prior to this admission? And what about the time window?  

Response: Thank you for your observations. We added the next paragraph in the manuscript, in order to clarify the issue:

“We included in the study only the patients who were in AF at the moment of inclusion in the study, respectively when the echocardiography was performed, independently if AF was diagnosed at inclusion or before. Most patients were already diagnosed with AF (permanent, persistent, or recurrent paroxysmal) before inclusion in the study. An ECG was performed at inclusion in all patients and 24 hours ECG Holter study was performed only in some patients with paroxysmal AF. In patients with persistent or permanent AF, 24-hours ECG Holter study was performed only in those with uncontrolled heart rate or in those suspected to associate other atrial or ventricular arrhythmias. Patients with a history of AF, who were in other cardiac rhythm at the initial assessment, were excluded from the study.”

Cordially yours,

Camelia Diaconu, MD, PhD, corresponding author